# Tactile Location Perception Encoded by Gamma-Band Power

**DOI:** 10.3390/bioengineering11040377

**Published:** 2024-04-15

**Authors:** Qi Chen, Yue Dong, Yan Gai

**Affiliations:** Biomedical Engineering, School of Science and Engineering, Saint Louis University, 3507 Lindell Blvd, St. Louis, MO 63103, USA; qi.chen@slu.edu (Q.C.); sophry.12@gmail.com (Y.D.)

**Keywords:** EEG, somatosensory, tactile, location perception, cushion, machine learning

## Abstract

Background: The perception of tactile-stimulation locations is an important function of the human somatosensory system during body movements and its interactions with the surroundings. Previous psychophysical and neurophysiological studies have focused on spatial location perception of the upper body. In this study, we recorded single-trial electroencephalography (EEG) responses evoked by four vibrotactile stimulators placed on the buttocks and thighs while the human subject was sitting in a chair with a cushion. Methods: Briefly, 14 human subjects were instructed to sit in a chair for a duration of 1 h or 1 h and 45 min. Two types of cushions were tested with each subject: a foam cushion and an air-cell-based cushion dedicated for wheelchair users to alleviate tissue stress. Vibrotactile stimulations were applied to the sitting interface at the beginning and end of the sitting period. Somatosensory-evoked potentials were obtained using a 32-channel EEG. An artificial neural net was used to predict the tactile locations based on the evoked EEG power. Results: We found that single-trial beta (13–30 Hz) and gamma (30–50 Hz) waves can best predict the tactor locations with an accuracy of up to 65%. Female subjects showed the highest performances, while males’ sensitivity tended to degrade after the sitting period. A three-way ANOVA analysis indicated that the air-cell cushion maintained location sensitivity better than the foam cushion. Conclusion: Our finding shows that tactile location information is encoded in EEG responses and provides insights on the fundamental mechanisms of the tactile system, as well as applications in brain–computer interfaces that rely on tactile stimulation.

## 1. Introduction

The perception of tactile-stimulation locations is an important function for our body to avoid danger and interact with the environment. Through perception, we gain knowledge on the object’s shape, orientation, temperature, etc. [1,2,3,4]. Unfortunately, previous studies on the perception of tactile locations have focused on the psychophysical aspects. For example, localization errors or precision have been measured with human behavioral tasks [5,6,7,8]. In those studies, tactile stimulation was usually applied to the upper body, such as the arms.

In rodents, decoding of tactile locations has been studied by applying stimulation to different whiskers or digits while recording from the primary somatosensory cortex [9,10,11]. Few studies have explored the possibility of decoding tactile locations using non-invasive electrophysiological methods in humans. One study using fMRI found that different brain areas are recruited in distance judgement, compared with contact judgement [12]. The most relevant study on decoding tactile locations using electrophysiological approaches was performed by Wang et al. [13]. They applied vibrations to four different locations on the right arm and recorded electroencephalography (EEG) from the human subjects. They found that the high-beta band (25–32 Hz) yielded the best decoding accuracy.

The present study generated vibrotactile stimulations to the epidermis of buttocks and thighs along the sitting interface while recording somatosensory-evoked potentials (SEPs) at the beginning and end of a sitting period. Compared with previous studies, we aimed to address three important aspects.

First, previous studies mostly examined the localization of tactile stimulation on the upper body, such as the arms, or both upper and lower bodies. Here, we attached the stimulators exclusively to the lower body, and particularly the part of tissue involved in sitting. Prolonged sitting is a known cause of cardiovascular diseases, diabetes, muscle deterioration, an increased enveloping fat layer, and other chronic diseases [14,15,16]. Sustained pressure and tissue load can have an accumulative effect on the seated buttocks, causing pressure ulcers to occur. Areas near bony prominences, namely the ischial tuberosity, are especially susceptible [14]. Unfortunately, few studies have systematically examined the effect of prolonged sitting on SEPs, and those studies focus on how SEPs vary with the sitting/semi-sitting posture [17,18,19,20]. In the present study, SEPs were obtained before and after the subject sat on a cushion for 1 h or longer to examine whether sitting had an effect on decoding tactile locations using SEPs.

Second, while examining whether sitting influences spatial-location perception, we altered the type of cushions the subject was sitting on. Conventional practice for tissue-stress alleviation is to use a soft and thick cushion, such as the thick foam cushion in this study, to allow adequate immersion and to distribute loads [21]. A better choice is “skin-protection cushions” designed to reduce pressure near bony prominences and accommodate orthopedic deformities. Clinically, those cushions have been shown to reduce ulcers by distributing load better than foam cushions [22]. Therefore, we anticipated to observe fewer changes in locational sensitivity after the human subject had sat on an air-cell cushion than on a foam cushion over the same period of time.

Last, the previous EEG study [13] did not examine information higher than 32 Hz. It has been known that gamma activity (>30 Hz, up to 200 Hz) contains information on important tactile features, such as texture [13], vibration frequency [23], multi-modal integration [24], and touch-pain perception [25]. Gamma somatosensory cortical oscillations are also found to be important for somatosensory gating during movements or pain perception [26,27].

Therefore, we performed the location classification using EEG signals separately extracted from the traditional delta, theta, alpha, beta, and gamma bands. An artificial neural net was applied to obtain classification accuracy on predicted tactile locations. The classification performance was examined for various experimental conditions, such as frequency bands, beginning and end of the sitting period, gender, and types of cushions.

## 2. Methods

### 2.1. Experimental Setup

Fourteen human subjects (aged 19–26), including six women and eight men, participated in the study. The subjects had no history of neuromuscular disorders. The experimental protocol was approved by the Institutional Review Board of Saint Louis University (Protocol #29002, entitled “Electroencephalography, sound detection, eye-tracking, and somatosensory-evoked potentials towards applications in prosthetics”). On each day, the subject sat in a chair that had either a flat and thick foam cushion (4–5″) or an air-cell cushion (ROHO High Profile, Permobil, Mt. Juliet, TN, USA; Figure 1A). The duration of the sitting period was either 1 h (Subjects S1 to S6) or 1 h 45 min (Subjects S7 to S14). During the sitting period, the subject stayed in the chair with the cushion and worked with a computer or a phone. He/she was allowed to go to the restroom when needed, but it rarely occurred.

A 2-D array of vibrators, called C-2 “tactors” (1¼-inch diameter, Figure 1B), were taped to the subject’s clothes around buttocks and thighs. Figure 1C shows the tactor placement. Tactors T1 and T2 were located near the left and right ischial tuberosities, respectively, where tissue load was supposed to be high during sitting. T3 and T4 were placed 3″ down from T1 and T2 on the thighs. The vibrations of the tactors were controlled by an 8-channel EAI Universal Controller (Engineering Acoustics Inc. Casselberry, FL, USA).

Stimulation waveforms were created with a Tactor Development Kit in MATLAB R2022b (Engineering Acoustics, Inc.). Each stimulation trial contained a 1-s, 25-Hz sinusoid followed by a 3-s silence. The gain of the vibrations was set to a medium value in the controller, which generated vigorous vibrations that could be clearly perceived even when the tactors were sat on. For each tactor location, 25 stimulations were applied consecutively. During each recording period, a total of 100 stimulations were created for the four tactor locations, taking an overall duration of 6.7 min. That is, the stimulation was applied to the subject at the beginning of the sitting period for 100 trials and again right after the sitting period ended for another 100 trials. Each subject completed three recording sessions for each experimental condition (e.g., type of cushion). Only one session was obtained from each subject on a given day. The cushion type alternated in consecutive sessions.

EEG signals were obtained with a 32-channel portable system (ANT Neuro, Hengelo, Netherlands) that comprised a head cap, an amplifier, and a Windows tablet computer. At the beginning of each stimulation, the computer sent a trigger via a StimTracker (Cedrus, San Pedro, CA, USA) to mark the event onset in the EEG amplifier. Figure 1D shows the 31 active electrodes, with the additional one, CPz, being the default reference electrode. During signal processing, the “common average montage” [28] was applied by subtracting the average from each of the 31 active electrodes.

### 2.2. Signal Processing and Classification

Signal pre-processing involved applying an Independent-Component-Analysis (ICA) approach [29] to remove blinking artifacts. The ICA algorithm identified the underlying independent brain-signal sources that had generated the recorded EEG so that eye-blinking and motion artifacts could be recognized and removed.

Signals were bandpass-filtered between 0.5 and 50 Hz using a 200th-order finite-impulse-response filter. Since each tactile stimulation lasted for 1 s, SEPs were extracted for the same period. Figure 2A,B shows representative SEPs obtained at the beginning and end of the sitting period (time 0 was the onset of vibrations) with Electrode CP1 for one subject. Each plotted SEP was an average over three days and, therefore, a total of 75 trials for each tactor location (color coded). As mentioned above, the SEPs have been bandpass filtered from 0.5 to 50 Hz.

Although there were notable differences in the 1-s SEP waveforms for different tactor locations (Figure 2A,B), during our preliminary testing, decoding the tactor locations based on single-trial SEP waveforms failed to yield significant detection results. Alternatively, we computed the power over traditional EEG frequency bands. Figure 2C,D shows periodograms for the same raw signals that yielded the SEPs in Figure 2A,B, but without any filtering. EEG waves were extracted according to the traditional bandwidths for delta (from 0.5 to 4 Hz), theta (from 4 to 7 Hz), alpha (from 8 to 12 Hz), beta (from 13 to 30 Hz), and gamma (from 30 to 50 Hz) [30]. Note that the gamma wave can go higher than 50 Hz. Since our EEG recordings contained decreased energy above 50 Hz and occasionally there could be interference from the 60 Hz electrical signals, we decided to stop at 50 Hz.

For a given frequency band, the power values of all 31 electrodes were used as the signal input (*x*) to the MATLAB built-in artificial neural net.
(1)$$x=\left[p_{1},\;p_{2},\ldots \;p_{31}\right],\;$$

Specifically, on each stimulation trial, the power of that frequency band was computed during the 1-s stimulation period, normalized by the entire power of all frequency bands (0.5–50 Hz), to form each $p_{i}$, with *i* being the channel number (*i* = 1, 2, … 31). Tactile location classifications were performed with the “fitnet” function provided by the MATLAB Machine-Learning Toolbox (R2022b). The neural net is a feedforward network. The input to the neural net had 31 normalized power values; that is, each input neuron was driven by one power value. The output had four artificial neurons, one representing each detected tactor location. There were five hidden neurons in between, and each perceptron has a sigmoid activation function [31] as below.
(2)$$f\left(x\right)=\frac{1}{1+e^{-a}}$$
where $a=\sum w_{i}p_{i}+b$, with $p_{i}$ being each power feature of the input. The training of the network model is to derive the best sets of $w_{i}$ (i.e., connection strengths between neurons) and *b* (i.e., the bias).

A total of 300 data trials (four tactor locations, each with 75 trials) were used in the training and validation of the neural net. A standard 10-fold cross-validation approach was adopted in evaluating the performance. Briefly, the 300 data trials were randomly split into 10 equal-number groups. Every time, one group was withheld as the test group, and the remaining nine groups formed the training pool to derive the weights and biases for the network. In the end, a classification accuracy, $\hat{y}$, was derived by averaging the performances of the ten test runs.

Next, to test the significance of classification performance, the 99% confidence interval [32], CI, was computed as
(3)$$CI=\pm 2.57\cdot \sqrt{\frac{\hat{y}(1-\hat{y})}{n}}$$

Here, *n* is the number of data points (n=75·4=300). The *CI* measures the degree of certainty in a sampling method. It is determined by the amount of variation and *n*. If the lower boundary, $\hat{y}-CI$, was greater than the chance level (i.e., 25%), statistical significance was reached, meaning that we are confident that the observation was real.

## 3. Results

### 3.1. Somatosensory-Evoked Potentials

Figure 2 shows example SEPs from a representative subject (S2) obtained with Electrodes CP1 at the beginning of a sitting period with the foam cushion. Here, each trace was an average of 75 trials, and different colors represent different tactor locations. A common observation for all the subjects is that sharp positive and negative peaks typically occur immediately after the stimulation onset and before 300 ms, with slow and negative activities starting at 400 ms. Although SEPs evoked by different tactor locations had slight variations in the SEP waveforms, classification of the tactor locations based on single-trial, single-electrode SEP waveforms yielded minimal discriminability (i.e., less than 30% correct rates, with the chance performance being 25% for four locations).

Next, we examined the spectral profiles of the recorded EEG (Figure 2C,D). Although there was a general trend of decreased energy with frequency, at certain frequencies, the spectra of the four tactor locations differed. Therefore, locational information may be encoded in the traditional EEG bands, which often vary with the electrode locations. Figure 3 shows general distributions of the power derived from entire recordings consisting of both stimulation and rest periods for all four locations. For each frequency band, the topography of three subjects was chosen to best represent the patterns we observed. The delta wave (0.5–4 Hz) was mostly found at the frontal and parietal lobes (Figure 3, 1st column). The theta (4–7 Hz) and alpha (8–12 Hz) waves can be focused at the frontal medial, parietal, or occipital lobes (Figure 3, 2nd and 3rd columns). The two highest-frequency bands, namely beta (13–30 Hz) and gamma (30–50 Hz), showed more sporadic distributions.

### 3.2. Decoding Tactile Locations

Once signals from different frequency bands have been extracted, the power of the signal during the 1-s tactile stimulation can be computed for each electrode. We then combined all the power values from 31 electrodes, forming a 31-dimentional vector as input to the artificial neural net.

Figure 4 shows the tactile-location decoding performance using signals from the five EEG bands for all the subjects sitting on the foam cushion. Error bars are 99% confidence intervals, and asterisks mark statistical significance when the lower boundary of the confidence interval was above chance (25%, the horizontal line). Generally speaking, delta and theta waves contained little spatial information (Figure 4A,B). The decoding performance was not much higher than chance (25% for four locations), even though occasionally the performance may reach the significance level. When the alpha wave was extracted (Figure 4C), most human subjects showed significant performance, with the highest accuracy being only 42%. With increasing frequencies, the beta and gamma waves showed even better results, reaching a 65% correct rate with the gamma waves for one subject (Figure 4D,E). One exception is that two subjects, S4 and S9, showed low or insignificant performance at the end of the sitting period.

Table 1 shows the three-way ANOVA test for the foam-cushion result presented in Figure 4. Three parameters, namely the frequency band (delta, alpha, …, etc.), sitting effect (before or after the sitting period), and gender effect, were examined. As expected, the frequency band had a dominant effect on the performance, yielding a *F* value of 77.42 and a *p* value of 0. Sitting had a significant effect (*F* = 7.94, *p* = 0.0056), meaning that after sitting for 1 h or 1 h and 45 min, the EEG responses had changed significantly. An unexpected finding is that gender also had a highly significant effect (*F* = 14.66, *p* = 0.0002). The exact gender effect will be examined later.

When the subjects sat on the air-cell cushion, the general observation remained the same. Spatial information became prominent with the alpha wave (Figure 5C) and reached the highest values with the beta and gamma waves (Figure 5D,E). A notable difference is that, with the gamma wave, the two subjects (S4 and S9) who showed significantly decreased performance after sitting on the foam cushion (Figure 4E) also showed significant performance after sitting on the air-cell cushion (Figure 5E).

Indeed, this observation agrees with the ANOVA result. Table 2 shows the three-way ANOVA test for the air-cell cushion, with the same three parameters as in Table 1. The frequency band was again a significant parameter (*F* = 77.47, *p* = 0). However, neither sitting nor gender caused significant changes in the EEG sensitivity to locations. Therefore, it is likely that the air-cell cushion maintained higher levels of sensitivity to location after the sitting period than the foam cushion, as expected.

To better demonstrate the sitting effect with different types of cushions and genders, the best classification performance over all five EEG bands for each subject was computed and plotted in Figure 6. With the foam cushion (Figure 6A), a few subjects who showed moderate performance at the beginning of sitting had notably declined performance after sitting. In contrast, the sensitivity values to tactile location were more tightly distributed along the diagonal. We have to admit that only a small number of subjects showed declining performance with the foam cushion.

Figure 6 also shows how gender may have influenced the results. There were a total of six females and eight males recruited for the study. For the foam cushion (Figure 6A), females generally showed the highest localization performances, which did not change much after the sitting period. In contrast, males’ performances were notably lower, with a tendency to decrease after the sitting period. That explains why gender played a significant effect on the foam-cushion condition (Table 1). With the air-cell cushion, females were still among the best performers (Figure 6B), although gender was no longer a significant factor (Table 2). This may be due to the limited number of human subjects, as we did not initially intend to study the gender effect.

Overall, to make a firm conclusion that the air-cell cushion better maintains the locational sensitivity on the buttocks and the EEG of females contains higher sensitivity to locations, a larger number of subjects will need to be recruited in future studies.

## 4. Discussion

### 4.1. Decoding Spatial Locations for Tactile Stimulation

The primary somatosensory cortex is located in the postcentral gyrus of the parietal lobe and projects to higher-order association cortices in the parietal lobe. It has been suggested that SEPs recorded with scalp electrodes are mediated by the posterior column-medial lemniscus pathway [33]. SEPs can be created either by vibrotactile stimulation on the epidermis or direct electrical stimulation of peripheral nerves [34]. Clinically, SEPs are widely used for intraoperative spinal-cord monitoring [35,36,37] or the diagnosis of neuropathies [17,38].

Positive and negative peaks can occur around 20, 30, 50, and 100 ms in SEPs generated by brief electrical stimulations to the median nerve in the upper limbs, with or without the accompanying vibrotactile stimulations [39,40,41]. Here, we applied sustained vibrations to the buttocks and thighs that lasted for 1 s on each trial. We observed multiple prominent peaks before 300 ms and a slow “recovery” afterwards until the end of the stimulation. Unfortunately, the temporal waveforms did not provide sufficient information regarding the tactor’s location (Figure 2), at least not on a signal-trial basis. Applying the artificial neural net to the temporal waveforms yielded a decoding performance of less than 30%. This is reasonable given the following facts about the evoked potentials.

First, the temporal waveforms of the SEPs are typically contaminated by large oscillations (namely the alpha, beta waves, etc.) that are unrelated to external stimulation. For that reason, averaging across many epochs is usually performed to derive a “clean” SEP [42], as shown in Figure 2 after averaging over 75 trials each. Indeed, when we combine multiple SEP trials and average those trials to form a single test sample, classification performance can be increased to >40% (not presented). In contrast, the band-power approach we tried next has the advantage of filtering out unwanted oscillations within a single trial.

Second, the tactor locations were on both the left and right sides of the lower body (Figure 1C). Because somatosensory information is mostly encoded by the contralateral side of the brain, we expect that multiple electrodes occupying both sides of the brain are necessary for decoding the exact stimulation location. Therefore, it is not surprising that single-electrode SEPs did not provide enough information on the tactile locations. In addition, the temporal waveform of a single SEP is already high-dimensional. If further combined across multiple electrodes, it would be too much for the artificial neural net to handle unless a large number of data trials were collected to train the classifier.

Alternatively, combining the signal power across all 31 electrodes yielded much better decoding performance, especially with the beta and gamma bands, for both types of cushions. We should point out the fact that the tactile stimulation frequency was 25 Hz. When we examined the frequency spectrum around 25 Hz, we could never identify a prominent peak with any electrode channel. Therefore, the observed classification performance was not due to EEG signals being synchronized to the tactile frequency. Rather, locational information seemed to be contained mostly in the two highest-frequency bands.

As mentioned earlier, the most relevant study on decoding tactile locations using EEG was performed by Wang et al. [13] achieved a classification accuracy as high as 96.76%. Our best performance (65%) was far less than that, with two possible explanations. First, they used a high-density 256-channel EEG system, whereas we only had 32 channels, including the reference electrode. Given that combinations across electrodes are critical in decoding spatial information, it is not surprising that their system yielded better performance than ours. Second, they examined four locations all on the right arm, whereas we examined four locations on the lower body, with two locations on each side. The difference in classification performance may be a result of different stimulation sites, but to confirm this hypothesis, the same stimulation and recording protocol (such as the high-density EEG system) will need to be applied while only the stimulation locations are varied.

### 4.2. Somatosensory Responses during Sitting

The human somatosensory system is composed of mechanoreceptors all over the body and parallel ascending pathways through the spinal cord. Mechanoreceptors specialize in delivering information on touch, pressure, vibration, and cutaneous tension [43]. Merkel’s disks in the epidermis are slowly adapting receptors that encode pressure. “Slowly adapting” means that their responses do not diminish with time. In contrast, Pacinian and Meissner corpuscles have relatively low detection thresholds (Pacinian lowest) [44] and are fast-adapting. Theoretically, they respond more at the beginning of sustained stimulations than steady state. Previous studies [45] found that cortical adaptation to sustained tactile stimulation, usually characterized as reduced peak amplitudes, can occur over seconds or minutes. In addition, sitting may deform the nerve endings and alter the ionic permeability of the receptor membranes of other types of mechanoreceptors.

Previous studies have explored how SEPs vary with the sitting position/posture [17,18,19,20]. In the study by Ormeci et al. [17], subjects were instructed to hold a fixed position for up to 20 min. They found an attenuated peak amplitude at the end of the sitting in response to electric stimulation of the pudendal nerve. The peaks occurred within 100 ms of the stimulation onset.

In the present study, human subjects were required to sit on a cushion for 1 h or 1 h and 45 min. Although, in the example shown in Figure 2A,B, the peak amplitudes of the SEPs decreased after sitting, this was not a consistent observation for all the subjects and/or electrodes. When we further examined the location classifications, the sensitivity did not always decrease after the sitting period. Nonetheless, certain subjects showed significantly degraded performance after sitting on the foam cushion but not on the air-cell cushion. The most convincing evidence is the three-way ANOVA test that showed “sitting” as a significant parameter for the foam cushion but not for the air-cell cushion. Given the fact that air-cell cushions are supposed to alleviate tissue stress from prolonged sitting, it is likely that we have discovered degraded sensitivity after sitting on the foam cushion. However, due to the fact that the number of subjects who notably decreased their sensitivity after sitting was small, we cannot draw a firm conclusion.

### 4.3. Gender Effect

This study did not initially plan to examine the gender effect on the decoding performance of tactile locations. When we analyzed the results in retrospect, we found that gender seemed to have made a difference, especially with the foam cushion. We found that female subjects generally showed the highest localization accuracy, and their performance did not degrade as much as that of male subjects after the sitting period.

Previous studies have shown that gender indeed matters in terms of the density of mechanoreceptors [46], tactile discrimination and acuity [47,48], etc. Generally speaking, young females tend to show better tactile perception. Given that the subjects recruited in the present study were college students, our EEG results may reveal something true regarding gender. However, a larger number of subjects will need to be tested in the future to validate this conclusion.

### 4.4. Applications

If the above speculation is true, this method may be used to evaluate the effectiveness of a certain type of cushion in alleviating stress. Currently, there are no direct measures of tissue stress and tactile sensitivity at the sitting interface–portions of the buttocks and thighs. Previous studies have measured seat pan pressures during prolonged sitting [49]. However, it has been shown that a simple contact-pressure measure is insufficient to reveal internal soft-tissue loads [50]. The pressure may remain the same over a long period of sitting, but the tissue may become less sensitive over time. Our finding may help develop a noninvasive electrophysiological approach to infer tissue sensitivity during prolonged sitting.

Our machine-learning results may also provide useful knowledge for brain–computer interfaces (BCIs) involving tactile stimulations. For patients who have impaired vision, BCIs based on visual stimulation are unsuitable. A recent study combining auditory and tactile stimulation to build a BCI speller achieved a much higher information transfer rate than using the auditory or tactile stimulation alone [51]. Another study built an electrotactile BCI based on SEPs [52]. Regarding the tactile stimulation protocol used in the present study, it is worthwhile to examine whether user attention can alter the classification performance to be used in tactile-based BCIs. This concept is hopeful because EEG responses generated by tactile imagery (e.g., imagined vibrotactile stimulation) have been shown to resemble the familiar EEG desynchronization of motor imagery [53].

### 4.5. Conclusions and Future Plans

This study explored EEG signal features that may encode locational information about tactile stimulations on the lower human body relevant to sitting. We found that, when combining signal powers from multiple electrodes that covered the entire scalp, relatively high-frequency bands, namely the beta and gamma waves, can best reveal locational information. The sensitivity to tactile locations encoded in this manner also seemed to degrade for some human subjects, which were exclusively male participants, after sitting on the foam cushion for 1 h or 1 h and 45 min, but not on the air-cell cushion. There seemed to be a gender effect in that females showed the highest sensitivity before and after the sitting period.

The conclusions should be treated with caution, as only a limited number of human subjects (i.e., 14) were tested as an exploratory effort. Those subjects already exhibited certain diversities, such as decreased sensitivity after the sitting period, even for the eight men alone. This may be caused by natural variations in the subjects’ physical conditions. In addition, previous studies have shown that EEG responses to vibrotactile stimulation may vary with age [54], whereas we only recorded from young college students. Future studies should examine our conclusions with a larger number of subjects while varying the parameters of age, gender, physical build, etc.

## Figures and Tables

**Figure 1 bioengineering-11-00377-f001:**
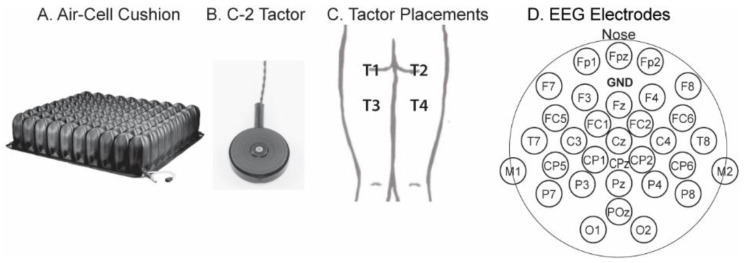
(**A**), a ROHO High Profile air-cell cushion. (**B**), a vibrational stimulator, “C-2 tactor”, with a diameter of 1¼ inches. (**C**), placement of the tactors, which were taped to the subject’s clothes. (**D**), electrode arrangement. GND, ground. CPz was the default reference electrode during recordings; the reference montage was later changed to the common average montage.

**Figure 2 bioengineering-11-00377-f002:**
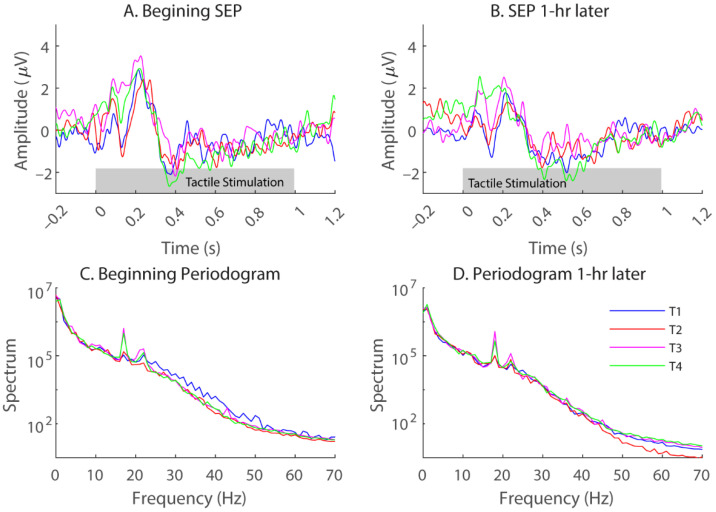
(**A**,**B**), representative somatosensory-evoked potentials (SEPs) obtained with Electrode CP1 from one subject (S2) at the beginning (**A**) and end (**B**) of the 1-h sitting period with the foam cushion. The SEPs were bandpass-filtered from 0.5 to 50 Hz. Different colors represent different tactor locations. Each trace is an average over 3 days (75 trials). Corresponding periodograms for the same SEPs are plotted in (**C**,**D**), but without any filtering.

**Figure 3 bioengineering-11-00377-f003:**
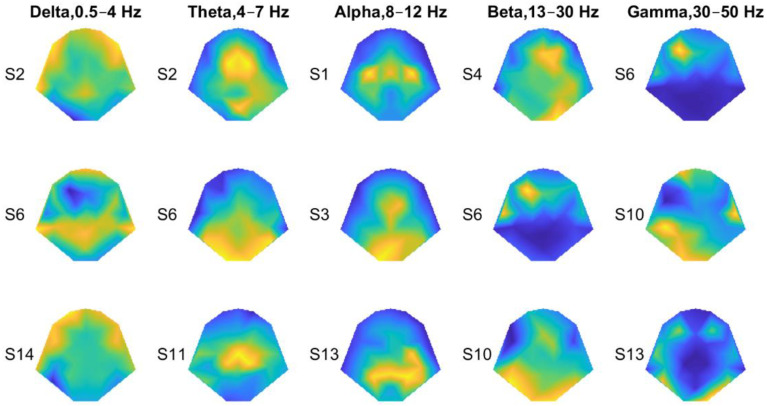
Representative power topographies for the traditional EEG frequency bands. Three representative topographies were chosen for each frequency band, with the subject IDs shown on the left. Those subjects were chosen to best present observed power patterns for all the subjects. The power was computed throughout each recording session, not just during the tactile-stimulation periods. Warm colors represent high power values. Cool colors represent low power values.

**Figure 4 bioengineering-11-00377-f004:**
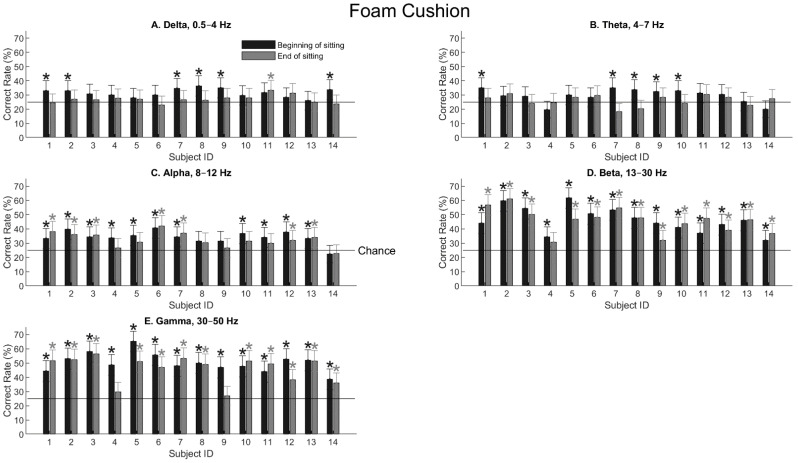
Tactile-location classification (i.e., the accuracy, $\hat{p}$) for all the subjects sitting on the foam cushion using power values of different EEG frequency bands. Dark and gray bars show the classification performance at the beginning and end of the sitting period, respectively. The first six subjects had a sitting duration of 1 h, whereas the rest eight subjects had a duration of 1 h 45 min. Error bars are 99% confidence intervals, and asterisks indicate statistical significance (*p* < 0.01) using the confidence-interval test (Equation (3)). Chance performance is 25% with four locations.

**Figure 5 bioengineering-11-00377-f005:**
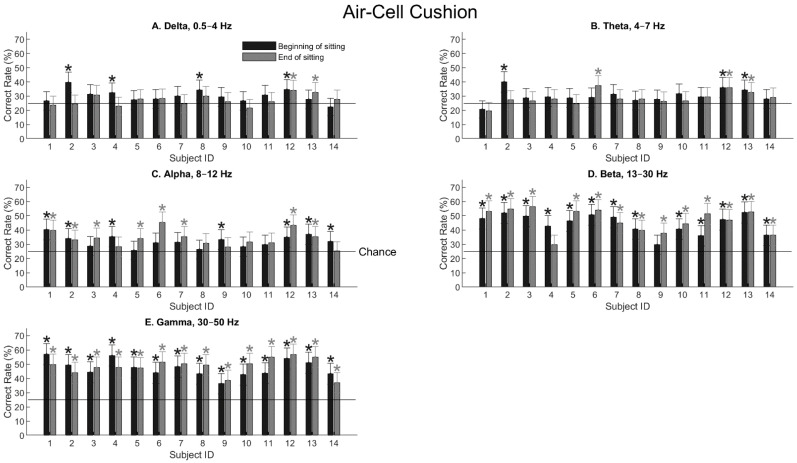
Tactile-location classification for all the subjects sitting on the air-cell cushion. The format is the same as in Figure 4. Asterisks indicate statistical significance (*p* < 0.01) using the confidence-interval test (Equation (3)).

**Figure 6 bioengineering-11-00377-f006:**
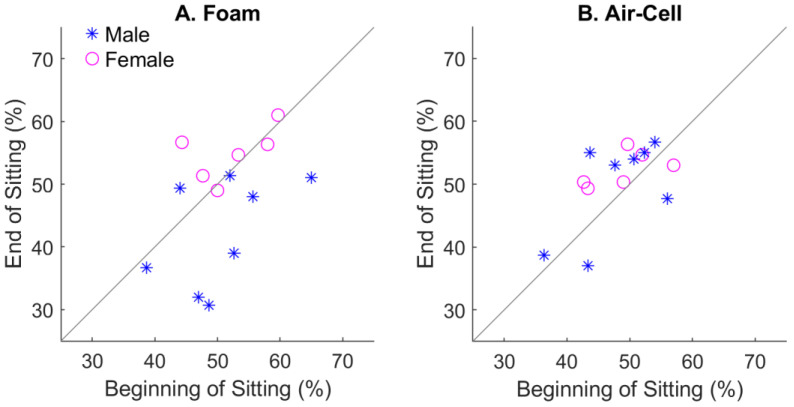
Effect of sitting on the classification performance for different types of cushions. (**A**), the foam cushion. (**B**), the air-cell-based cushion. Each symbol represents a human subject, with gender separated.

**Table 1 bioengineering-11-00377-t001:** Three-way ANOVA test for the classification performance with the foam cushion as a parameter of frequency band, gender, and sitting effect.

Source	Sum Sq.	df	Mean Sq.	*F*	Prob > *F*
Sitting	269.1	1	269.07	7.94	0.0056
Frequency	10,496.3	4	2624.08	77.42	0
Gender	496.9	1	496.9	14.66	0.0002
Sitting:Frequency	78.6	4	19.64	0.58	0.6781
Sitting:Gender	18	1	17.95	0.53	0.4681
Frequency:Gender	316.6	4	79.15	2.34	0.0592
Error	4202.8	124	33.89		
Total	15,650.8	139			

**Table 2 bioengineering-11-00377-t002:** Three-way ANOVA test for the classification performance with the air-cell cushion as a parameter of frequency band, gender, and sitting effect.

Source	Sum Sq.	df	Mean Sq.	*F*	Prob > *F*
Sitting	0.5	1	0.55	0.02	0.8937
Frequency	9462.5	4	2365.63	77.47	0
Gender	2.6	1	2.62	0.09	0.7701
Sitting:Frequency	152.5	4	38.12	1.25	0.294
Sitting:Gender	24.7	1	24.69	0.81	0.3703
Frequency:Gender	137	4	34.24	1.12	0.3496
Error	3786.2	124	30.53		
Total	13,542.9	139			

## Data Availability

The data presented in this study are available on request from the corresponding author. The data are not publicly available due to ethical requirements.

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
