# Peer review of "Tactile Location Perception Encoded by Gamma-Band Power"

_bioengineering, 2024, doi:10.3390/bioengineering11040377_

Round 1

Reviewer 1 Report

Comments and Suggestions for Authors

The paper is well written it concerns tacticle location perception by gamma band power during EEG. Included are 6 figures and 1 table no major grammer/spelling errors detected. 

The study is about EEG, specifically - Tactile Location Perception Encoded by Gamma-Band Power 

- aim to characterize spatial location perception of the upper body.

- Characterization of EEG location perception

- I am not familiar with the related literature, but I think that it adds its conclusions (that  fundamental mechanisms of the tactile system, as well as applications in brain-computer 23 interface that rely on tactile stimulation. 

- potentially additional human subjects (other species/ages).

major issues  - provide ethics statement. Include gender/age for the human samples. Potentially apply additional statistical tests on the data (e.g. such as  ANOVA with FDR)     minor issues - none found.         

Author Response

We have addressed all the questions, including adding the gender information and changing the two-way ANOVA to a three-way ANOVA to examine gender effect, see attached file for details.

Reviewer 2 Report

Comments and Suggestions for Authors

(1)The details of signal processing and classification are not enough. Is there a math model or math equation method to show?

(2)What is the physical meaning of CI? The p and n in CI equation,what's meaning?

(3)What is your conclusion and future research plan? It should be added a new section to illustrate.

(4)The nearest new references are less.

Author Response

Thanks for the useful suggestions. We have addressed all the questions, see the attached file for details.
